# WASP family proteins regulate the mobility of the B cell receptor during signaling activation

Ivan Rey-Suarez[1,2], Brittany A. Wheatley[3,4], Peter Koo[5], Anshuman Bhanja[6], Zhou Shu[6,7], Simon Mochrie[8], Wenxia Song[6], Hari Shroff[2] & Arpita Upadhyaya[1,3,4]*

Regulation of membrane receptor mobility tunes cellular response to external signals, such as in binding of B cell receptors (BCR) to antigen, which initiates signaling. However, whether BCR signaling is regulated by BCR mobility, and what factors mediate this regulation, are not well understood. Here we use single molecule imaging to examine BCR movement during signaling activation and a novel machine learning method to classify BCR trajectories into distinct diffusive states. Inhibition of actin dynamics downstream of the actin nucleating factors, Arp2/3 and formin, decreases BCR mobility. Constitutive loss or acute inhibition of the Arp2/3 regulator, N-WASP, which is associated with enhanced signaling, increases the proportion of BCR trajectories with lower diffusivity. Furthermore, loss of N-WASP reduces the diffusivity of CD19, a stimulatory co-receptor, but not that of FcγRIIB, an inhibitory co-receptor. Our results implicate a dynamic actin network in fine-tuning receptor mobility and receptor-ligand interactions for modulating B cell signaling.

[1] Biophysics Program, University of Maryland, College Park, MD 20742, USA. [2] National Institute of Biomedical Imaging and Bioengineering, National Institutes of Health, Bethesda, MD, USA. [3] Department of Physics, University of Maryland, College Park, MD 20742, USA. [4] Institute for Physical Science and Technology, University of Maryland, College Park, MD 20742, USA. [5] Department of Molecular & Cellular Biology, Harvard University, Cambridge, MA 02138, USA. [6] Department of Cell Biology & Molecular Genetics, University of Maryland, College Park, MD 20742, USA. [7] Division of Immunology, Children's Hospital of Chongqing Medical University, Chongqing, China. [8] Department of Physics, Yale University, New Haven, CT 06520, USA. *email: arpitau@umd.edu

B cells are an important component of the adaptive immune system. B cells sense antigen using specialized receptors known as B cell receptors (BCRs) that trigger signaling cascades and actin remodeling upon binding antigen on the surface of antigen-presenting cells (APC)[1–3]. Signaling activation results in spreading of the B cell on the APC surface leading to the formation of a contact zone known as the immunological synapse[4]. Antigen crosslinking aggregates BCRs in lipid rafts, enabling lipid raft-resident Src kinase to phosphorylate their immunoreceptor tyrosine-based activation motifs (ITAMs)[5–7]. Signaling BCRs assemble into microclusters, which grow via movement of BCRs and their incorporation into these microclusters[1,8]. BCR clustering is dependent on the probability of receptor–receptor interactions at the plasma membrane[9], and is in part dictated by the lateral mobility of receptors. Thus, elucidating the mechanisms that regulate BCR movement in the cell membrane is critical for understanding BCR signaling.

The cortical actin network in cells is known to form juxta-membrane compartments that can transiently confine the lateral movement of membrane proteins[10–12], including BCRs in B cells[13]. Treanor et al.[13] showed that in unstimulated B cells, inhibition of actin polymerization leads to an increase in lateral diffusivity of BCR and is accompanied by signaling that is reminiscent of activation. The transient dephosphorylation of ezrin and actin depolymerization induced by BCR–antigen interaction results in the detachment of the cortical actin from the plasma membrane concurrent with a transient increase in the lateral movement of surface BCRs[14]. Activation of Toll-like receptors sensitizes BCR signaling, by increasing BCR diffusivity through the remodeling of actin by cofilin, an actin binding protein that disassembles actin filaments[15]. The submembrane actin cytoskeleton also modulates the concentration of inhibitory co-receptors[16,17] in the vicinity of BCR microclusters, thereby ensuring the rapid inhibition of activated BCRs.

A consensus picture that emerges from these studies is that in resting B cells, the actin network serves as a structural barrier for BCRs, regulating their mobility by steric interactions. Beyond this structural role, considerable evidence points to a role for dynamic actin in regulating BCR signaling and activation. Our previous work has shown that inhibition of actin polymerization by low concentrations of Latrunculin A following antigen stimulation inhibits the growth of BCR microclusters[18], suggesting that actin dynamics plays a direct role in modulating BCR mobility. Furthermore, actin regulatory proteins are known to regulate signaling and have been implicated in the control of microcluster formation. Wiskott–Aldrich Syndrome protein (WASP) and Neural-WASP (N-WASP) are scaffold proteins which are activated downstream of BCR activation. They link receptor signaling to actin dynamics through the activation of the Arp2/3 complex to promote the growth of branched actin networks[19–21]. Liu et al.[22] found that N-WASP plays an important role in the deactivation or attenuation of BCR signaling. B cells from N-WASP conditional knockout mice exhibit delayed cell contraction and sustained signaling compared with control cells. However, these studies have largely focused on changes in the dynamics of cell spreading and BCR microcluster movement and coalescence on a global cell-wide scale. Whether and how actin dynamics directly modulate nanoscale BCR diffusion and signaling and the role of actin regulatory proteins in this process remain open questions[23].

The involvement of actin regulatory proteins in modulating BCR signaling suggests that actin dynamics may directly affect nanoscale BCR diffusion. We show that inhibition of actin nucleators or deletion of WASP family proteins leads to an overall lower diffusivity of BCR and its signaling co-receptor, CD19. Concomitantly, we find that inhibition of WASP family proteins also reduces actin flows, suggesting that the effect of WASP family proteins or downstream effectors on BCR mobility and signaling is actin mediated. Our findings reveal a role for actin dynamics in modulating nanoscale receptor diffusion, highlighting the importance of the dynamic actin network in regulating receptor mobility and signaling.

## Results

**B cell receptor motion spans a wide range of diffusivity.** Primary murine B cells were allowed to spread on a supported lipid bilayer coated with mono-biotinylated Fab' fragment of BCR-specific antibody (mbFab) that induces BCR signaling. We used interference reflection microscopy (IRM) to visualize the spreading and contraction of B cells on supported lipid bilayers (Fig. 1a, top panels), and total internal reflection fluorescence (TIRF) microscopy to analyze the clustering of BCR and coalescence of BCR clusters during cell contraction (Fig. 1a, bottom panels). B cell receptor diffusivity was extracted from single-molecule imaging of BCR by TIRF imaging (Fig. 1b–d). We verified that B cells underwent signaling activation in our experimental conditions by labeling with phosphotyrosine as well by quantifying the increase in intensity of BCR clusters due to coalescence (Supplementary Fig. 1). To label the BCR, Alexa Fluor (AF) 546 labeled mbFab was added to the imaging medium at low concentrations (<1 μM) so that only single molecules were detected[24]. Cells were imaged from the moment they contacted the bilayer and time-lapse movies of single BCR molecules were recorded in TIRF (Fig. 1c). Single molecules detected in each frame and identified to be within the cell contour were localized with high precision (~20 nm) and linked frame by frame to generate tracks[25]. A representative compilation of the tracks obtained from a single cell during a 10-min period is shown in Fig. 1d. The tracks are color coded for short-time diffusivity, calculated using the covariance-based estimation method[26]. The estimator is unbiased and does not need a regression analysis to estimate diffusion coefficients, making it ideal for obtaining diffusion coefficients from short single-particle trajectories. The cumulative distribution of diffusion coefficients measured across the population of tracks shows variation in diffusivity over several orders of magnitude. These results suggest that BCR exhibit a wide spectrum of diffusivities, which may be indicative of their signaling properties and their biochemical state. Moreover, we found a larger proportion of BCR with higher mobility in the first minute compared with later time points (Fig. 1e), consistent with the onset of signaling[27]. This is reflected in the comparison of the diffusivity distributions measured at the indicated time points which shows that the diffusivity at minute 1 was significantly higher than at subsequent time points (Fig. 1f). In contrast, B cells in contact with transferrin-tethered bilayers (non-activating control) exhibited an overall higher BCR diffusivity throughout the imaging period and did not show a progressive reduction over time (Supplementary Fig. 2a).

**Identification of distinct diffusive states for BCR.** In order to obtain a better understanding of the diffusive properties of BCR, we employed a systems level classification algorithm, perturbation-expectation maximization (pEM), which uses machine learning to extract the set of distinct diffusive states that best describes a diffusivity distribution[28,29]. The premise underlying pEM is that various biochemical interactions of a protein lead to a finite number of distinct diffusive behaviors (diffusive states). pEM determines the number of diffusive states in an unsupervised, statistically correct fashion using the Bayesian Information Criterion (see "Methods" for a more detailed rationale). We used pEM-v2, which accounts for non-normal

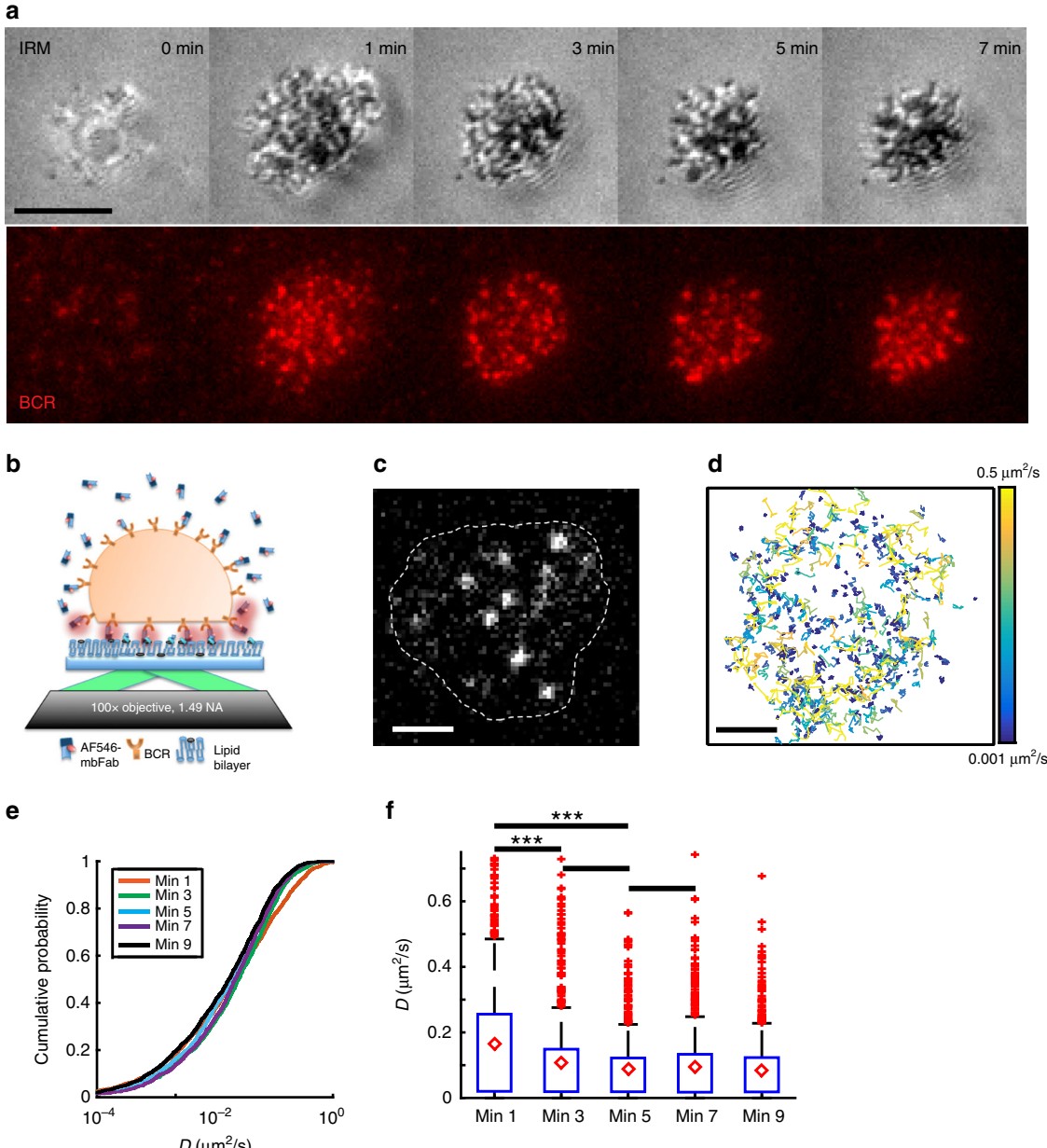

**Fig. 1 Single-particle tracking reveals wide range of BCR mobility. a** Panel showing primary murine B cell spreading (IRM, above) and BCR clustering (TIRF, below). Scale bar is 3 µm. **b** Experimental schematic, indicating activated murine primary B cells, placed on supported lipid bilayers coated with mono-biotinylated fragments of antibody (mbFab). Cells are imaged in TIRF mode and the concentration of AF546 labeled mbFab is kept low enough to image single-molecule events. **c** Representative TIRF image with the bright dots representing single BCR molecules. The cell contour is obtained from an IRM image taken after TIRF imaging. Scale bar is 1 µm. **d** The collection of tracks obtained for a control cell during a 10-min period imaged at 33 Hz for 1000 s every minute. The tracks are color coded for diffusivity. Scale bar is 1 µm. **e** Cumulative distribution function (CDF) for the diffusivities measured at 1, 3, 5, 7, and 9 min after activation for BCR in B cells from control mice. **f** Boxplot showing BCR diffusivities at the indicated time points ($N = 15$ cells). The mean is marked with red diamonds, the bottom line represents the lower quartile, the upper line the upper quartile, the whiskers show the extent of the rest of the data, and red crosses are the outliers. Significance of differences was tested using the Kruskal–Wallis test (***$p < 0.001$; 1 min vs 3 min, $p = 0.0008$; min 1 vs min 5, $p = 0.000038$; min 3 vs min 5, $p = 0.1767$; min 5 vs min 7, $p = 0.8614$).

diffusive modes and the high heterogeneity of the cell membrane by splitting trajectories into shorter segments and identifying transitions between different diffusive states across segments. All single-molecule trajectories were split into 15-frame segments and the classification analysis was performed on the set of all of these 15-frame-long tracks. pEM analysis of all BCR trajectories from B cells identified eight distinct states, revealing a far greater complexity of diffusive behavior than is apparent from approaches that average over all tracks or that impose two diffusive

states only[15]. Fig. 2a shows representative trajectories assigned to each state.

In control cells, all eight states displayed simple diffusion, as shown by the ensemble mean-square displacement (eMSD) plots (Fig. 2b). For all states, the representative mean diffusivity was preserved over time (Fig. 2c). For each time point, we calculated the fraction of tracks assigned to each state—the population fraction (Fig. 2d). The population fraction of the fastest moving state, State 8, rapidly decreases within the first few minutes and

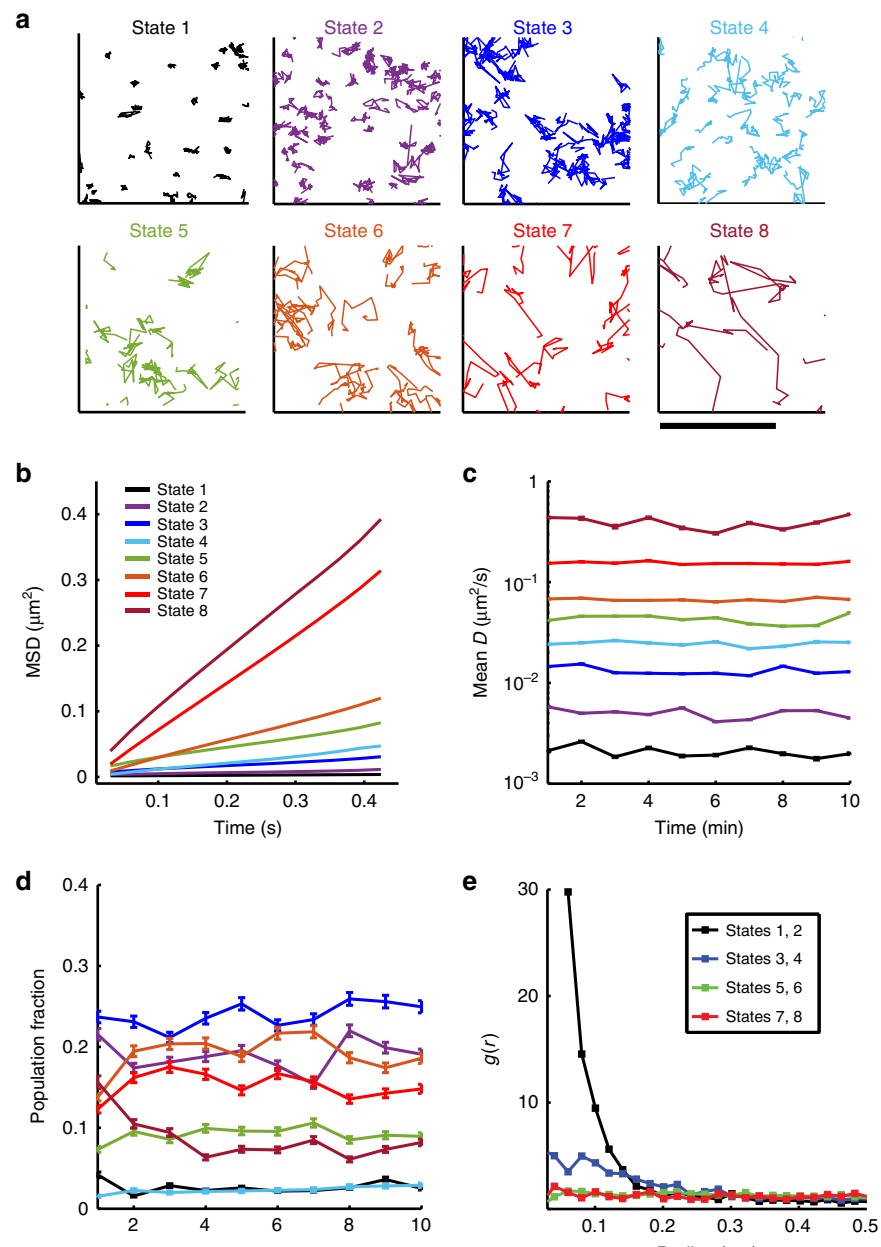

**Fig. 2 Perturbation expectation maximization analysis identifies eight distinct diffusive states for BCR in control cells. a** Characteristic tracks belonging to each of the BCR diffusive states identified by pEM. Diffusivity increases from State 1 (slowest) to State 8 (fastest). Scale bar is 1 μm. **b** Ensemble mean-square displacement (eMSD) plots for each of the states. Colors corresponding to different states are as shown in the legend. ($N = 15$ cells). **c** Plot showing the mean diffusivity for the trajectories belonging to each state at every time point. Error bars represent the standard error of the mean. **d** Plots showing the fraction of BCR tracks that are sorted in each state at every time point. Error bars represent a confidence interval of 95% on the population fraction calculation. **e** Plot of pair correlation as a function of distance for all states. Source data are provided as a Source Data file.

then stabilizes. This corresponds to the decrease in diffusivity for the higher mobility fraction (Fig. 1e). The population fractions of the other states fluctuate over time but do not show a clear trend. To validate that these low-mobility states resulted from BCR activation, we analyzed single-molecule BCR trajectories for cells on transferrin-tethered bilayers. For these cells there was a near complete loss of BCR trajectories in the lowest mobility states (States 1 and 2), while the population fractions of the fast mobility states (States 5–8) were higher than those in B cells interacting with lipid bilayers through the BCR (Supplementary Fig. 2b, c).

To quantify the spatial distribution of trajectories, the first position of each nonredundant trajectory (for each distinct particle) was used to compute the spatial pair correlation function as a

function of distance, $g(r)$, between the localized spots (Fig. 2e)[30]. $g(r)$ measures the normalized probability of finding a second localized fluorophore at a given distance, $r$, from an average localized fluorophore. A value of 1 indicates that receptors that occupy a given state are randomly organized, whereas values > 1 denote a higher probability of finding receptors in a given state at shorter distances, indicative of clustering. The range $r$ over which $g(r) > 1$ denotes the scale of clustering. To calculate pair correlation functions, we combined trajectories belonging to pairs of states that are closest in diffusivity (e.g., States 1 and 2; States 3 and 4, and so on). The lowest mobility states, States 1 and 2, display $g(r)$ that is significantly larger than 1 for small values of $r$ (Fig. 2e), suggesting that these trajectories are significantly more densely clustered

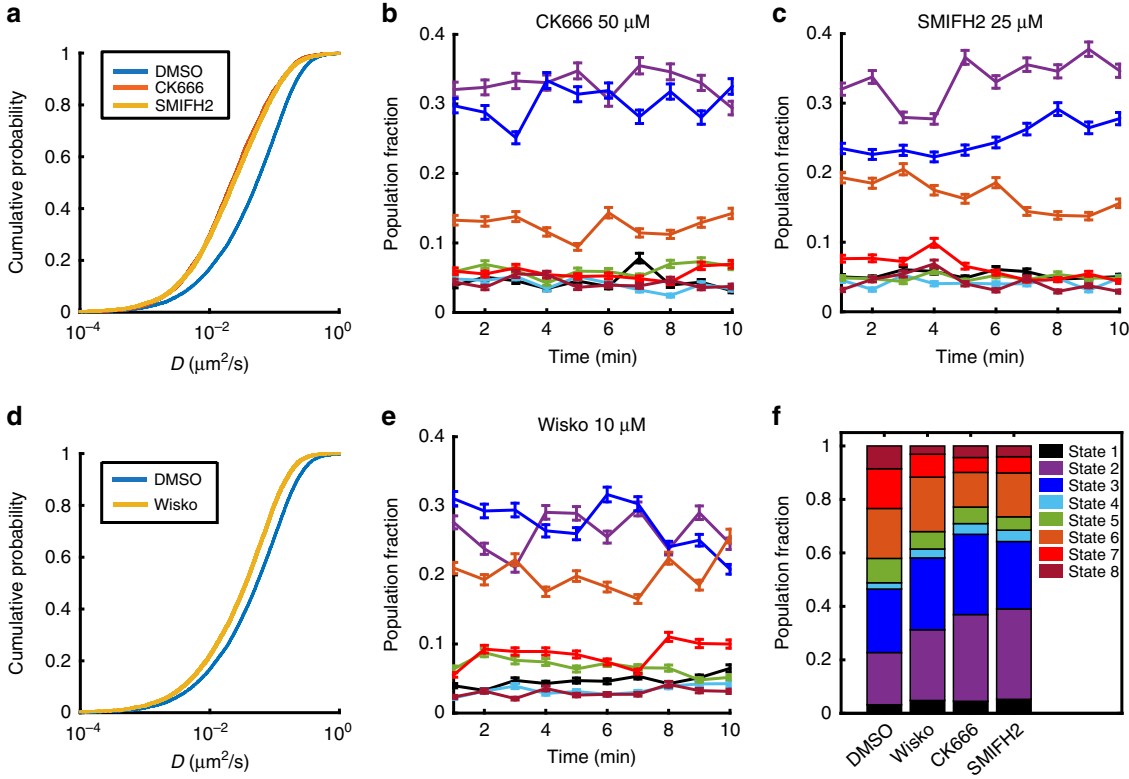

**Fig. 3 Inhibition of actin nucleation decreases BCR diffusivity. a** Plots of BCR diffusivity distributions for cells treated with CK666 (inhibitor of Arp2/3 complex) or SMIFH2 (inhibitor of formins). ($p < 0.001$, Kruskal–Wallis test for comparison between DMSO and CK666, or DMSO and SMIFH2). **b** Population fraction over time for cells treated with CK666. **c** Population fraction over time for cells treated with SMIFH2. The colors corresponding to the different states are as shown in **f. d** BCR diffusivity distribution for cells treated with wiskostatin (Wisko) compared with DMSO control. ($p < 0.01$, Kruskal–Wallis test for comparison between DMSO and Wisko) **e** Population fraction over time for cells treated with wiskostatin. Error bars in **b**, **c**, and **e** represent a confidence interval of 95% on the population fraction calculation. **f** Overall distribution of population fractions for cells treated with wiskostatin, CK666 and SMIFH2 (Number of cells: DMSO, $N = 14$; Wisko, $N = 11$; CK666, $N = 10$; SMIFH2, $N = 16$). Source data are provided as a Source Data file.

compared with other states. States 3 and 4 show low clustering, while the other higher mobility states display a largely homogeneous distribution. Of note, the slowest diffusive states, States 1 and 2, appear to be the ones that correspond to BCR in clusters.

**Actin-nucleating proteins regulate BCR mobility**. In order to investigate how BCR diffusivity is modulated by actin dynamics, we inhibited the two dominant actin-nucleating pathways. Addition of CK666, a small molecule inhibitor of the Arp2/3 complex results in decreased mobility of surface BCRs as compared with DMSO-control cells (Fig. 3a). Inhibition of formin, an actin-nucleating protein that polymerizes bundled actin, using SMIFH2 results in BCR with lower mobility as compared with control cells (Fig. 3a). The reduction in overall BCR diffusivity by formin inhibition is similar to that by Arp2/3 inhibition. pEM analysis was performed on the set of BCR tracks from cells treated with these inhibitors. The low-mobility states, States 2 and 3, contribute to over 60% of all BCR trajectories in B cells treated with CK666, compared with 40% in control cells (Fig. 3b, f). SMIFH2-treated cells show a slightly different behavior (Fig. 3c, f), wherein only State 2 displays an overall increase (35% of all trajectories) relative to controls (20% of all trajectories). The growth of branched actin networks by Arp2/3 requires its activation by the WASP family proteins. We next asked how these actin regulators modulate BCR diffusion by treatment with wiskostatin, an inhibitor of WASP family regulators. We found that application of wiskostatin results in a decrease in BCR diffusivity (Fig. 3d) and an increase in the population fraction of BCRs in States 1 and 2 (Fig. 3e, f). Overall, inhibition of actin-nucleating

proteins, Arp2/3 and formin, as well as upstream regulators reduces BCR diffusivity, while increasing the population fraction of the slow diffusive states as compared with control cells. These results collectively implicate actin dynamics in maintaining the heterogeneity of BCR mobility and nanoscale organization.

**WASP family proteins modulate B cell receptor diffusivity**. B cells, like all immune cells, express the hematopoietic-specific Arp2/3 regulator, WASP as well as the ubiquitous N-WASP. These two proteins have high homology and share many overlapping functions but have distinct effects on B cell spreading, BCR signaling, and microcluster formation[22]. To test how WASP family regulators affect nanoscale BCR diffusion, we utilized B-cell-specific conditional N-WASP KO mice (cNKO) and WASP KO mice (WKO). We used the same single-molecule imaging strategy to obtain tracks of single BCR over time as shown for cNKO cells in Fig. 4a. The different track colors correspond to tracks with different diffusivity and show a preponderance of tracks with lower diffusivity (blue) as compared with BCR tracks in control cells. However, the decrease in diffusivity for the high-mobility tracks in the first few minutes that was observed for control cells is not evident in cNKO cells (Fig. 4b).

To determine whether the differences in BCR diffusivities between control and cNKO cells are related to the known differences in BCR signaling in these two conditions, we used pEM analysis to assign BCR trajectories to diffusive states in cNKO B cells. The data containing the trajectories of BCR from control and cNKO cells was analyzed together and eight distinct states were identified, as for the control case. The

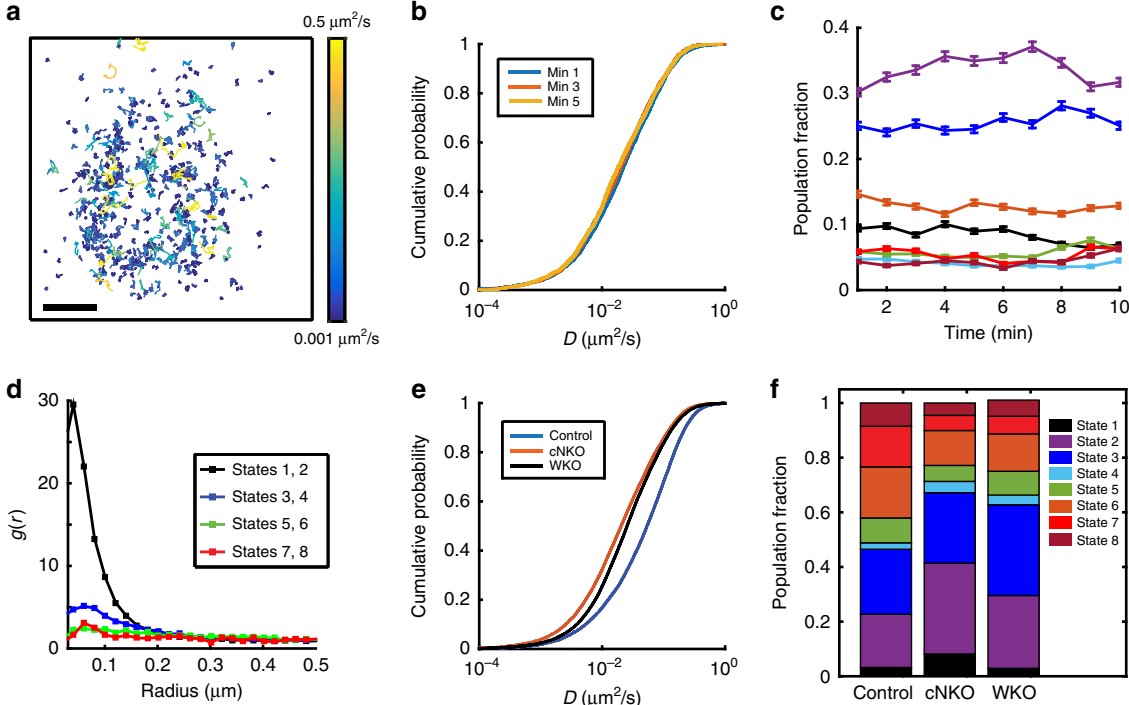

**Fig. 4 N-WASP knockout leads to predominance of BCR molecules in lower mobility diffusive states. a** Collection of tracks obtained from pEM analysis of BCR molecules in a cNKO cell during a 10-min period. The tracks are color coded for diffusivity. Scale bar is 1 μm. **b** Cumulative distribution function for diffusivities measured at 1, 3, and 5 min after BCR stimulation in B cells from cNKO mice. **c** Plots of population fractions of eight distinct diffusive states as a function of time for BCR in cNKO cells. Error bars represent a confidence interval of 95% on the population fraction calculation. The colors corresponding to the different states are as shown in **f**. **d** Pair correlation function plots of the trajectories in different diffusive states for cNKO cells. **e** The distribution of diffusivities from the 5–10 min time period after activation, for BCR in control, WKO, and cNKO cells. The distributions for control are significantly different from WKO and cNKO cells (control cells $N = 15$, WKO cells $N = 21$ cells, $p < 0.001$ and cNKO cells $N = 17$, $p < 0.0001$ Kruskal–Wallis test). **f** Comparative population fractions for BCR in different states over the entire time period in control, WKO, and cNKO cells. Significance levels for the differences are in Supplementary Table 1. Source data are provided as a Source Data file.

population fraction of BCR in each state remains relatively constant over time (Fig. 4c), and the eMSD of each state displays simple diffusion (Supplementary Fig. 3). Pair correlation analysis shows that in cNKO cells, States 1 and 2 display clustering similar to control cells (Fig. 4d). States 3 and 4 show a somewhat nonhomogeneous distribution, while the higher mobility states have a homogenous distribution.

Given that the diffusivity remains largely unchanged for times beyond 5 min in both types of cells, we compared the CDF of diffusion coefficients measured for times between 5 and 10 min for control and cNKO cells. We find that BCR in cNKO cells display much slower diffusivity than control cells (Fig. 4e), as also evident from the preponderance of tracks with lower diffusivity (blue tracks, in Fig. 4a, compared with Fig. 1d). Similar to control B cells, the lower mobility states, States 1–3 are the most dominant states in cNKO cells. Comparison of population fractions (pooled across time) of occupied states showed significant differences (see Supplementary Table 1) between control and cNKO cells, with a significantly larger proportion of trajectories occupying the lowest mobility states (States 1–3) in cNKO cells (Fig. 4f).

We next examined how WASP regulates nanoscale BCR mobility, by single-molecule imaging of BCR in B cells from WKO mice to obtain BCR trajectories (Supplementary Fig. 4a). We found that WASP deletion results in a reduction of BCR diffusivity (Fig. 4e). pEM analysis of BCR tracks in WKO B cells again led to the identification of eight distinct states similar to control and cNKO cells, with the population fraction of each state relatively constant over time (Supplementary Fig. 4b). Pair

correlation analysis of tracks again showed States 1 and 2 as being the most clustered (Supplementary Fig. 4c). However, pEM analysis revealed qualitatively significant differences in the diffusive states between WKO and cNKO B cells. We found that WKO B cells had similar population fractions of States 1 and 2 as control B cells, unlike cNKO B cells (Fig. 4f). This indicates that WASP and N-WASP have differential effects on the mobility and putative signaling states of BCR. Taken together, the predominance of lowest mobility states of BCRs in activated B cells and the increase in these lowest mobility states in cNKO B cells, which have been shown to have higher levels of BCR signaling than control B cells, are consistent with a model in which the lower diffusivity of BCR corresponds to its signaling, clustering and activation state.

In addition to identifying diffusive states, pEM analysis can determine transitions between these states along individual trajectories (Supplementary Fig. 5a). We selected longer tracks (>30 frames) and identified the state(s) to which the subtracks had been assigned and quantified the frequency of transitions from a given state to different states (Supplementary Fig. 5b, c). BCRs tend to remain in their current state or switch to an adjacent one. BCRs in the three slowest diffusive states were the most stable in both control and cNKO cells, showing the least transition probability to other states. These observations suggest that fast diffusive particles are more likely to encounter a cluster and be incorporated into it, thereby transitioning into the neighboring slower state. BCRs in cNKO cells tend to transition towards slow diffusive states, especially States 2 and 3 (Supplementary Fig. 5c). Particles in fast diffusive states are less

stable and transition into slower states more frequently in cNKO than in control cells. This is consistent with the higher population fraction of BCR in slow diffusive states in cNKO cells.

**N-WASP modulates the diffusivity of the co-receptor CD19**. To better understand the nature of the BCR diffusive states that were enhanced in cNKO activated cells, we investigated how N-WASP affects the diffusivity of CD19, a stimulatory co-receptor[31]. CD19 is recruited to the BCR upon antigen binding, enhancing BCR activation. A previous study using super-resolution imaging found that in resting B cells, CD19 resides in nanoclusters separated from IgM BCR nanoclusters, while in activated B cells, CD19 and BCR nanoclusters are colocalized[32,33]. Thus, single-molecule studies of CD19 have the potential to reveal additional insight into signaling BCR states. We used instant Structured Illumination Microscopy (iSIM)[34] which enables super-resolution imaging with a lateral resolution of 145 nm and an axial resolution of 350 nm to simultaneously image CD19 and BCRs. Consistent with previous reports, CD19 and BCR microclusters colocalized to within our resolution limit of 140–150 nm in activated B cells (Fig. 5a, b, Supplementary Fig. 6a). Furthermore, these microclusters moved together towards the center of the contact zone (Supplementary Fig. 6b, Supplementary Movie 1). To identify the diffusive states of CD19, we performed single-molecule imaging of CD19 using AF594 labeled anti-CD19 antibody at low-labeling concentrations and with the same methods as for imaging BCR. Analysis of CD19 tracks in control cells during the 10-min imaging period (Fig. 5c), shows that the diffusivity of CD19 is lower than that of BCR, consistent with the abundance of short tracks (blue) (compared with Fig. 1d). pEM analysis of CD19 tracks again resulted in eight distinct states with mean diffusivities preserved over time (Fig. 5d). Pair correlation analysis of CD19 molecules in both control and cNKO cells shows higher correlations in States 1 through 4 at shorter distances than the other states, indicative of a clustered configuration (Fig. 5e, f). As observed for BCRs, trajectories from States 1 and 2 of CD19 show the highest degree of clustering at short distances, while the faster moving states show a more homogeneous distribution.

Interestingly, the cumulative distribution plots of the diffusivities showed that CD19 diffusion in cNKO cells is significantly lower than in control cells (Fig. 5g). The diffusivities of the eight states found for CD19 are very similar to those found for BCR, allowing us to compare the population fractions between these receptors (Fig. 5h). States 1, 2, 4, and 8 are more predominant for CD19 while States 3, 6, and 7 are more populated for BCR in control cells. The population fraction of the lowest mobility states (States 1 and 2) for CD19 show a significant increase in cNKO cells compared with control cells (Fig. 5i), as observed for BCR. These results suggest that knocking out N-WASP affects the diffusivity of BCRs and CD19 in a similar way, slowing down their overall mobility and likely maintaining their interactions inside signaling clusters. We next examined the roles of different actin-nucleating factors and regulatory proteins on CD19 diffusivity using inhibitors. All inhibitors reduced overall CD19 diffusivity as compared with control (Supplementary Fig. 6c). For all cases, the population fraction of States 1, 2, 3, and 6 showed a significant increase (Supplementary Fig. 6d). These data suggest that BCR and CD19 in States 1 and 2, which are also enhanced in cNKO cells, are likely to be in a signaling state.

**N-WASP KO has a limited effect on FcγRIIB diffusivity**. To determine whether actin-mediated modulation of mobility is specific to the BCR and its stimulatory co-receptor CD19 or reflects a more general change in the diffusive properties of the membrane environment due to changes in the cortical actin network, we tested whether the mobility of FcγRIIB, an inhibitory co-receptor of the BCR, is similarly affected by the lack of N-WASP. The FcγRIIB receptor is a transmembrane receptor expressed in B cells and inhibits BCR signaling and BCR clustering by the recruitment of phosphatases such as SHIP (SH2-domain containing inositol polyphosphate 5′ phosphatase)[35,36], when it is colligated with the BCR by antibody–antigen immune complexes. FcγRIIB is known to exhibit relatively high diffusivity as compared with BCRs in quiescent B cells and its mobility is altered by mutations associated with autoimmune diseases[17]. In the absence of colligation, the mobility of FcγRIIB has the potential to yield important insight into the generic diffusion properties of transmembrane receptors.

We studied the diffusivity of FcγRIIB (without colligation) in activated B cells using the same methods used for the other receptors. Figure 6a shows a compilation of FcγRIIB tracks in a control cell over a 10-min period. In contrast to the significant slowdown of BCR diffusivity in cNKO cells, FcγRIIB diffusivity is minimally affected (Fig. 6b). pEM analysis of the trajectories showed seven states with stable mean diffusivities (Fig. 6c). As with the other receptors studied so far, pair correlation analysis of FcγRIIB shows that States 1 and 2 display signs of clustering, States 3 and 5 also display clustering but to a lesser degree, while all other states display a more homogeneous distribution (Fig. 6d, e). Moreover, inhibition of actin nucleators (Arp2/3 and formins) or WASP family proteins did not alter FcγRIIB mobility (Supplementary Fig. 7a–c) nor change the population fraction of BCR trajectories in different diffusive states (Supplementary Fig. 7d). The lack of drastic changes in FcγRIIB mobility in the absence of N-WASP or inhibitors of actin nucleation compared with BCR and CD19 suggests that N-WASP-mediated regulation of receptor mobility is specific to the BCR.

**WASP family proteins regulate actin dynamics in B cells**. To test whether the changes in BCR mobility and nanoscale organization induced by inhibition of actin regulators are associated with alterations in actin dynamics, we imaged primary B cells from Lifeact-EGFP transgenic mice activated on supported lipid bilayers. We used iSIM to obtain high spatial resolution images amenable to quantitative analysis of actin flows to determine the effect of inhibitors of actin nucleation and upstream regulators. In primary B cells, the actin network is organized into highly dynamic foci (indicated by blue arrows) and a thin lamellipodial region at the cell periphery (indicated by yellow arrows) in both untreated and wiskostatin-treated cells (Fig. 7a, Supplementary Movie 2). We used spatio-temporal image correlation spectroscopy (STICS)[37] to quantify the speed and directionality of actin flows (Fig. 7b). From the actin flow velocity vector maps, we generated heat maps showing actin flow speeds and directions relative to the cell center. Actin flow speeds do not display any systematic spatial dependence in either wiskostatin-treated or control cells (Fig. 7c). However, wiskostatin-treated cells display a significant reduction in actin flow speed (Fig. 7d), suggesting that upstream regulators of Arp2/3 are required for generating a dynamic actin network.

To determine the directionality of actin flow vectors, we defined a directional coherence measure as the cosine of the angle relative to a vector pointing to the centroid of the cell. Flows towards the cell center have value 1 and flows away from the center have value −1 with other angles spanning intermediate values within this range. Spatial maps of directional coherence values reveal that actin flows are not spatially correlated over large regions for either wiskostatin-treated cells or control cells (Fig. 7e). Probability density functions (PDF) of the directional

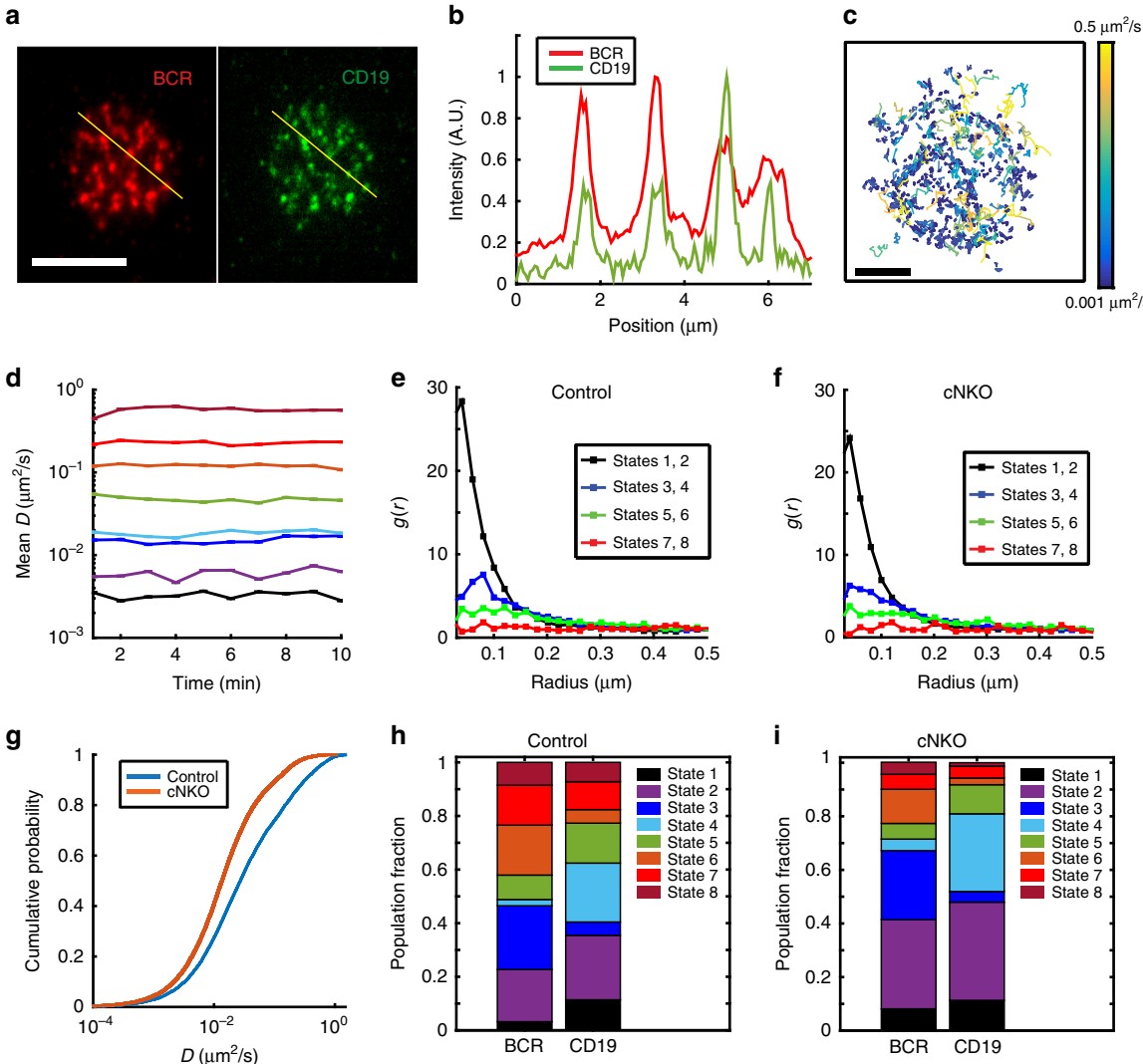

**Fig. 5 N-WASP expression modulates CD19 diffusivity. a** Instant SIM images of activated B cell, showing that AF546 labeled BCR (red) and AF488 labeled CD19 (green) reside in clusters that colocalize to within the ~150 nm resolution limit. Scale bar is 5 µm. **b** Intensity profiles for BCR (red) and CD19 (green) fluorescence along the yellow lines as drawn in **a**. **c** Compilation of CD19 tracks over a 10-min period in an activated control B cell. Scale bar is 1 µm. **d** Plot showing the mean diffusivity of each of the eight diffusive states obtained from pEM analysis as a function of time. The colors corresponding to the different states are as shown in **h**. Error bars represent the standard error of the mean. **e, f** Pair correlation function plot for all states for control and N-WASP-KO cells respectively. **g** Cumulative Probability distribution of diffusivities showing that mobility of CD19 in cNKO cells is significantly lower than in control cells (control cells $N = 10$, cNKO cells $N = 11$, $p = 0.0013$ Kruskal–Wallis test). **h, i** Comparison of population fractions of BCR and CD19 in different states for control and cNKO cells, respectively. Source data are provided as a Source Data file.

coherence values over the entire contact zone show that during the early stage of activation (0–5 min), actin flows are predominantly directed either inwards or outwards (relative to the cell centroid) with no significant difference between control and wiskostatin-treated cells (Fig. 7f). However, during the late stage of activation (5–10 min), wiskostatin-treated cells display greater inward actin flows towards the cell center and less outward flows compared with control cells (Fig. 7g).

In order to more directly examine the effect of the actin nucleation inhibitors Arp2/3 and formin, we used iSIM to visualize actin dynamics in cells treated with CK666 and SMIFH2 (Supplementary Fig. 8a). The actin organization and dynamics appear to be qualitatively different in inhibitor-treated cells, with the presence of more linear bundle-like structures in CK666-treated cells and the apparent loss of these structures in SMIFH2-treated cells (Supplementary Fig. 8a). STICS analysis shows that actin flow speeds are significantly decreased in both CK666 and

SMIFH2-treated cells compared with control cells (Supplementary Fig. 8b).

## Discussion

Here, we used single-molecule imaging and a novel machine learning based analysis method to obtain a better understanding of the diffusive properties of BCR during activation. These methods allow us to classify single-molecule trajectories into states with distinct diffusivities and correlate BCR diffusivity states with potential signaling states. Activation results in a reduction of BCR mobility with a larger fraction of BCRs in low-mobility diffusive states, suggesting that signaling BCRs have low diffusivity. Moreover, BCRs in states with low diffusivities display a greater degree of spatial clustering. Inhibition of actin-nucleating proteins reduces both BCR diffusivity and actin flows, suggesting that the reduced BCR mobility is actin

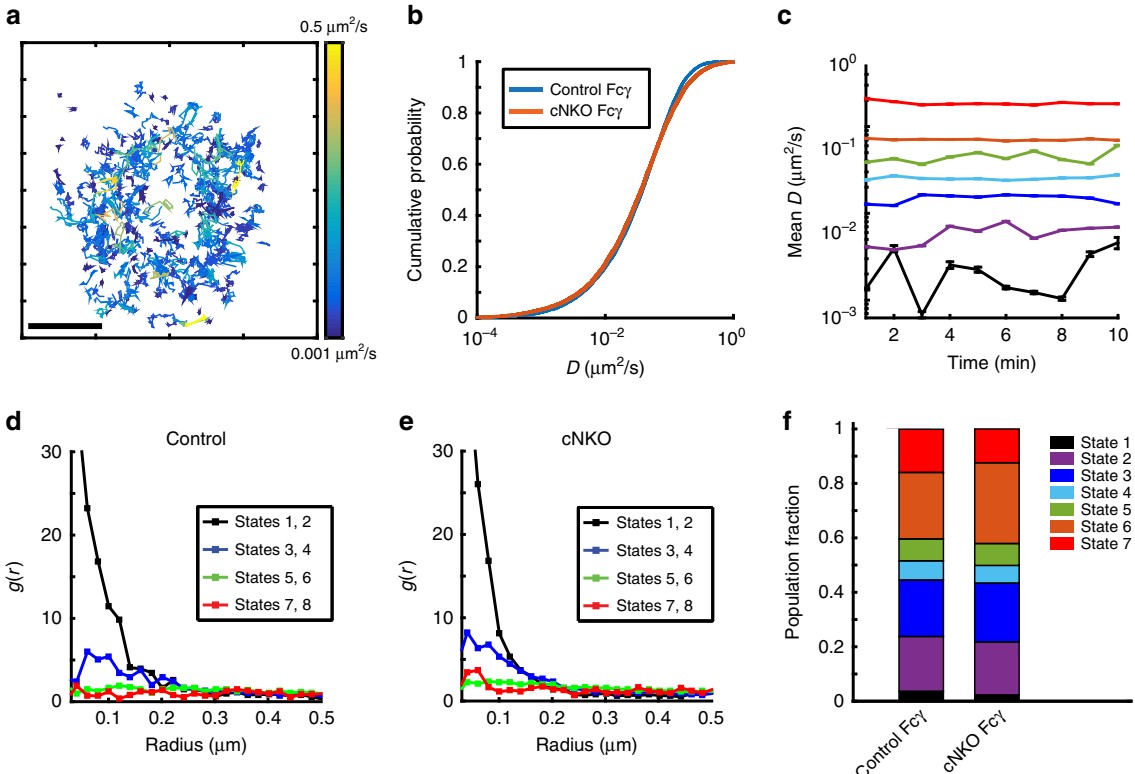

**Fig. 6 FcγRIIB mobility is mildly affected by the lack of N-WASP. a** A set of single-molecule tracks of FcγRIIB from an activated B cell over a 10-min period. Scale bar is 1 μm. **b** Cumulative distribution plots for diffusivity of FcγRIIB molecules in control and cNKO cells (control cells $N = 12$, cNKO cells $N = 12$). **c** pEM analysis of single FcγRIIB molecule trajectories uncovered seven states. Plot shows the mean diffusivity of each state at every time point. Error bars represent the standard error of the mean. The colors corresponding to the different states are as shown in **f**. **d, e** Pair correlation function plot for all states in control and cNKO states respectively. **f** Comparison of the population fraction of different diffusive states of FcγRIIB in control and cNKO cells. Source data are provided as a Source Data file.

mediated. Consistently, loss of WASP and N-WASP, upstream activators of actin dynamics, also leads to decreased BCR diffusivity compared with control cells. To further relate BCR signaling and diffusivity, we took advantage of the fact that B cells from cNKO mice[22] display enhanced signaling. We found an increase in the fraction of low-mobility states in cNKO B cells. Further, the stimulatory co-receptor, CD19[31,38], shows similar reduction of mobility and enhancement of low-mobility states in cNKO B cells, suggesting that low diffusivity states correspond to signaling states. In contrast, the inhibitory co-receptor, FcγRIIB[39], showed no difference in diffusivity or population fraction across mobility states between control and cNKO B cells, suggesting that the effect of N-WASP on BCR and CD19 mobility is not global for all membrane proteins during B cell activation. Overall, our study reveals the link between BCR diffusion and signaling and suggests that actin dynamics, mediated by WASP family proteins, regulate BCR signaling by modulating the diffusivity of BCR and its co-receptors and their nanoscale organization during activation.

Inhibition of formins resulted in reduced BCR mobility, but with somewhat different effects on the diffusive states as compared with the effects of Arp2/3 inhibition. This could be due to direct reductions of linear actin structures generated by formin or by alterations of branched actin networks produced by the cooperation of Arp2/3 and Diaphanous formins. Studies have shown that formation of branched filaments by Arp2/3 requires an existing actin filament and a nucleation-promoting factor to start polymerizing actin[40]. The formin, mDia1, can generate mother filaments that set the basis for the formation of the actin meshwork, which underlies the formation of different types of cell

protrusions. Such cooperative interactions have been observed between the formin mDia1 and Arp2/3 for the generation of lamellipodia and ruffles[41] and the formation of proper actin architecture during invasive podosome formation[42]. Furthermore, T cells from mDia1-knockout mice cannot form lamellipodia or ruffles, and exhibit defective cell motility[43]. In summary, our studies suggest that optimal BCR signaling requires homeostatic balance between actin networks generated by multiple actin-nucleating proteins.

Extending previous studies showing that N-WASP knockout is associated with enhanced signaling[22], our observations suggest that low-mobility BCR trajectories are associated with signaling states. Using both pair correlation and pEM analysis, we showed that the low diffusivity of BCRs on the surface of cNKO B cells is accompanied by a lower diffusivity of its stimulatory co-receptor CD19, strengthening our hypothesis that signaling states of BCRs correlate with those that display decreased mobility. Stone et al. showed that the spatial positions of the BCR and Lyn, a signaling kinase, become correlated after antigen stimulation, and this correlation is accompanied by a reduction in diffusivity of both molecules[44]. Using a photoactivatable antigen, Wang et al.[27] showed that BCR diffusivity decreased following antigenic stimulation. During activation, conformational changes of ITAM-containing receptors, changes in local lipid environment and interactions with other proteins may alter the mobility of membrane receptors. For instance, stimulation of mast cells through FcεRI receptor crosslinking, induces the clustering of FcεRI and a concomitant reduction of its diffusivity that depends on the average number of receptors in the cluster[45]. The clustering of FcεRI receptor is accompanied by the redistribution of the

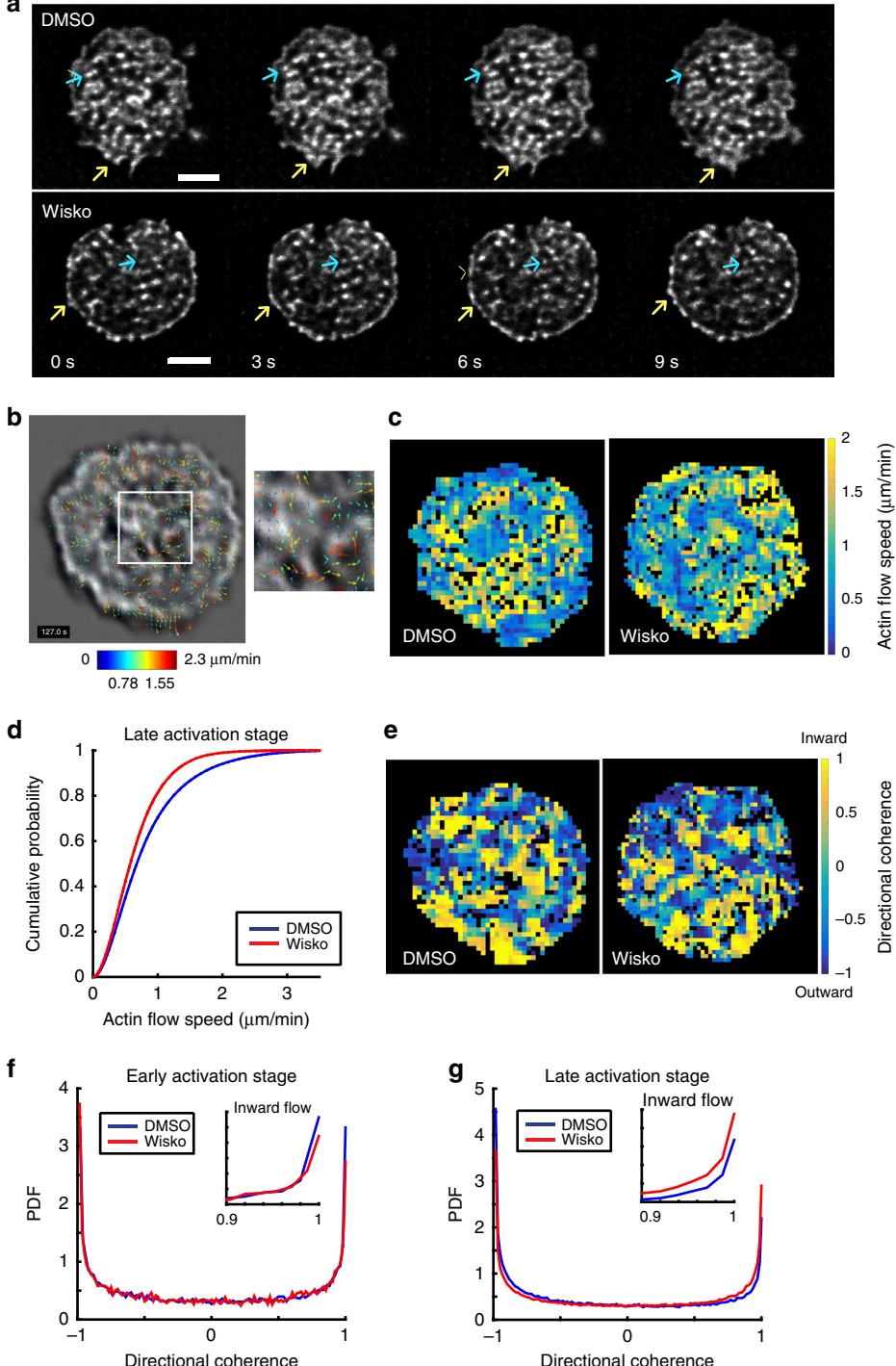

**Fig. 7 Effect of actin regulators on actin dynamics in activated B cells. a** iSIM images of activated Lifeact-EGFP B cells at consecutive time points for two conditions: DMSO carrier-control and wiskostatin (10 μM concentration). The initial time corresponds to 5 min after spreading initiation. The blue arrows in the images indicate the emergence of actin foci and the yellow arrows point to spreading and contraction of the lamellipodial region of the cells. Scale bars are 2 μm. **b** STICS (Spatio-temporal image correlation spectroscopy) vector map showing actin flows represented by velocity vectors indicating flow direction and color coded for flow speed. In the zoomed region, the velocity vectors show the flow direction and flow speed. The vector map is overlaid on top of a grayscale image of Lifeact-EGFP. **c** Pseudocolor map of actin flow speeds corresponding to 2 min after cell spreading for representative DMSO-control and wiskostatin-treated cells. **d** Cumulative probability distribution of actin flow speeds for DMSO-control cells (blue, $N = 11$ cells) and cells treated with wiskostatin (red, $N = 12$ cells). ($p = 0.00074$, Kruskal–Wallis test). **e** Directional coherence maps indicating the flow directions, which ranged from inward (1) to outward (−1). **f, g** Probability density function plots showing directional coherence values of actin flow in cells during the early stage of activation (**f**) or cells in the late stage of activation (**g**), with subplots highlighting the flow fraction defined as inward flow (see Methods). During early stages, the fraction of inward flow is 0.143 for DMSO and 0.1425 for wiskostatin-treated cells ($p = 0.5188$—not significant); during late stages, the fraction of inward flow is 0.113 for DMSO and 0.1518 for wiskostatin-treated cells ($p < 0.001$).

signaling proteins Lyn kinase and Syk kinase into clusters[46]. Thus, our findings are broadly consistent with prior studies that link receptor diffusivity and clustering with their signaling state.

A well-accepted model of the regulation of BCR diffusion by the actin cytoskeleton in resting B cells posits that the actin network imposes diffusional barriers on BCR and other receptors and signaling proteins[13,14]. Activation leads to the dissolution of these barriers either by severing of actin filaments[15] or removal of cytoskeletal/membrane anchors[14], thereby enhancing BCR diffusion, leading to further activation. However, previous models have not considered an active role for actin dynamics in signal regulation. Based on our observations, we suggest that in contrast to prior models, the role of the actin cytoskeleton in BCR signaling goes beyond providing a mechanical barrier for receptor diffusion. Specifically, we propose that nonequilibrium, rapidly changing actin flows may serve to stir the cytoplasm adjacent to the membrane, thus changing the reaction environment of receptors and signaling molecules, and modulating the reaction rates in the juxtamembrane regions of the cytoplasm[47]. Actin regulatory proteins modulate the level of nonequilibrium actin dynamics and thereby alter receptor mobility and their signaling states.

According to our new model, we propose the following dynamics of early BCR signaling. At rest, BCRs are confined within membrane compartments defined by the actin cytoskeleton (Fig. 8a). Early BCR signaling leads to the loss of these barriers, as well as increased actin dynamics (Fig. 8b). Our imaging studies have revealed that B cell actin dynamics is not characterized by spatially coherent directional flows. Rather, actin dynamics are highly complex, with sharp changes in speed and directionality, both spatially and temporally. This dynamics may also be associated with the formation of nonequilibrium actin structures such as asters and foci[48,49]. This active stirring, combined with the release of BCRs from diffusion traps, may drive receptors into clusters and facilitate receptor interactions with activating kinases[23]. At later stages, further increases in actin

dynamics and outward actin flows could decrease reaction rates between BCRs and activating kinases or increase reaction rates between BCRs and inhibitory co-receptors, making signaling states more unstable, facilitating downregulation of signaling (Fig. 8c, top). Inhibition or loss of upstream regulators of actin nucleation results in a reduction of actin dynamics (Fig. 8c, bottom). This may decrease actin-mediated mixing of BCRs in the membrane, likely enabling BCR to enter and remain in signaling states (clusters) leading to enhanced signaling and preponderance of low-mobility states (or conversely increased interactions with phosphatases leading to signaling inhibition).

Based on our observations, we suggest that actin dynamics in the cell may be used to fine-tune the levels of signaling activation. Modulation of the structure and dynamics of actin networks, by changing the expression levels or spatial distribution of actin regulatory proteins, may provide the cell with a powerful way to regulate signaling over rapid timescales. These properties are likely to be a general feature of cells in the immune system whose function depends on rapid response to external stimuli, and illustrate general principles of immune receptor signaling.

## Methods

**Mice and cell preparation**. B-cell-specific N-WASP knockout (CD19^Cre/+ N-WASP^Flox/Flox, cNKO) mice and littermate control mice (CD19^+/+ N-WASP^Flox/Flox) were generated by breeding N-WASP^Flox/Flox (CD19^+/+ N-WASP^Flox/Flox) mice on a 129Sv background, and CD19^Cre/+ mice on a C57BL/6 background[22]. WASP knockout (WASP^−/− CD19^+/+ N-WASP^Flox/Flox, WKO) mice were generated by breeding N-WASP^Flox/Flox littermate control (CD19^+/+ N-WASP^Flox/Flox) mice on a mixed (129Sv and C57BL/6) background, and WASP knockout (WASP^−/− CD19^+/+) mice on a 129Sv background. Transgenic Lifeact-EGFP mice on a C57BL/6 background were obtained from the Wedlich-Söldner lab[50]. Mice selected for experiments were between 2 and 4 months old with no gender preference. Naive primary B cells were isolated from mouse spleens using a negative selection procedure as described before[24]. After extraction, cells were kept at 4 °C and cell aliquots were prewarmed at 37 °C for 5 min before being added to the bilayer. All experiments involving animals have been approved by the University of Maryland Institution Animal Care and Usage Committee.

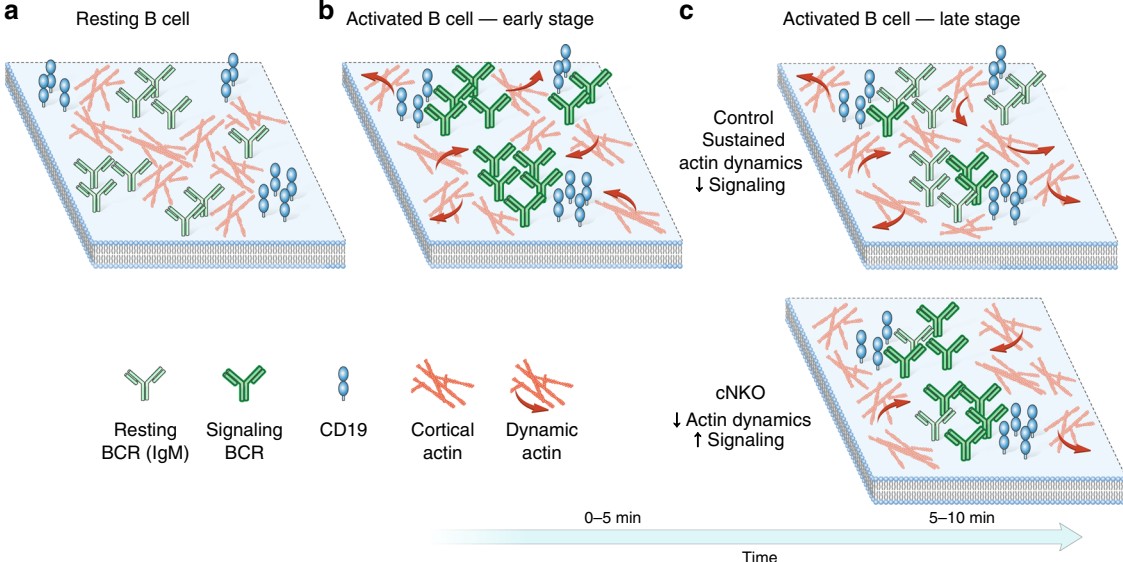

**Fig. 8 The actin cytoskeleton regulates B cell receptor mobility and signaling in different stages.** Representative cartoon showing receptor distributions on a section of the B cell membrane: **a** Resting B cell membrane: actin networks restrict receptor lateral movement and interactions. **b** B cell membrane at the early signaling activation stage. Actin remodeling enhances receptor mobility allowing for interactions between receptors, specifically BCR and CD19, enhancing signaling. Actin flows towards the center and edges of the immune synapse in similar proportions. **c** B cell membrane at later activation stages. Top: actin flows stir the cytoplasm at the membrane vicinity, increasing the mixing of receptors in the membrane and thereby allowing signal inhibitory molecules to downregulate BCR signaling. Bottom: N-WASP knockout reduces actin dynamics and changes the balance of actin flow directionality at later stages (5–10 min) of activation, leading to enhanced signaling.

**Fluorescent antibodies and inhibitors**. For inhibition of formin and Arp2/3, cells were incubated with inhibitors for 5 min at 37 °C before being added to the imaging chamber, which had the inhibitor at the same concentration used for incubation. SMIFH2 (Sigma-Aldrich) was used at a 25 μM concentration. Arp2/3 complex inhibitor I, CK666 (Calbiochem) was used at 50 μM. For N-WASP inhibition, wiskostatin B (EMD Bioscience) was used at 10 μM to incubate cells for 1 h at 37 °C. Mono-biotinylated fragment of antibody (mbFab′-anti-Ig) was generated from the F(ab′)$_2$ fragment (Jackson Immuno Research, West Grove PA) using a published protocol[51]. FcγRIIB (CD32) antibody (Cat# 553141, BD Biosciences) was conjugated with Alexa Fluor 546 using Molecular Probes Protein labeling kits (Cat# A10237, Invitrogen) following manufacturer protocols. For labeling of CD19 we used the Alexa Fluor 594 anti-mouse CD19 antibody at 0.15 μg/ml (Cat# 115552, BioLegend).

**Sample preparation for single-particle tracking**. Glass slides were kept in Nanostrip (Cyantek) overnight and then rinsed with dd-H$_2$O and dried with filtered air. Supported lipid bilayers were prepared by incubating slides with 10 μM DOPC/DOPE-cap-biotin liposome solution for 10 min at room temperature. The slides were rinsed with filtered PBS (1×) and then incubated for 10 min with 1 μg/ml solution of streptavidin. Slides were rinsed again with PBS and then incubated with unlabeled mono-biotinylated fragment of antibody (mbFab) solution at 18 μg/ml. PBS was replaced with L-15 (CO$_2$ independent media with 2% FBS) before imaging. 0.75 μl of 0.05 mg/ml AF546 labeled mbFab was added to a 250 μl volume of media in the imaging chamber.

As non-activating controls, streptavidin coated lipid bilayers on coverslips were incubated with biotinylated transferrin (Jackson) (16 μg/ml) for 10 min at room temperature. Slides were rinsed with PBS and then the PBS was replaced with L-15 (CO$_2$ independent media with 2% FBS) before imaging. Cells were added to transferrin-coated lipid bilayers and incubated for 5 min. For single-particle imaging, 0.75 μl of 0.05 mg/ml AF546-labeled mbFab was added to a 250 μl volume of media in the imaging chamber.

**Microscopy**. For single-molecule imaging of BCR we used an inverted microscope (Nikon TE2000 PFS) equipped with a 1.49 numerical aperture 100× lens for TIRF imaging and an electron multiplying charge coupled device (EMCCD) camera (iXon 897, Andor). In order to image single molecules on the cell membrane for extended periods of time we added a low concentration of the fluorescent antibody in solution as shown in Fig. 1. Cells were imaged from the moment they contacted the bilayer and time-lapse movies were recorded for a 30 s duration every minute (1000 frames acquired at 33 Hz). Figure 1b shows a representative frame where the cell outline was obtained from an IRM image taken after the single-molecule movie. The molecules detected at each frame were localized with high precision (~20 nm) and linked frame by frame to create tracks[29] using a MATLAB routine. Taking into account motion blur, pEM estimates localization precision to range between 20 nm (slowest states) and 80 nm (fastest state). Imaging of Lifeact-EGFP expressing murine primary cells spreading on supported lipid bilayers was performed using iSIM[34], with a 1.42 numerical aperture 60× lens (Olympus), a 488 nm laser for excitation with 200 ms exposure times and a PCO Edge camera. Images obtained were post processed with background subtraction and deconvolution. The final lateral resolution for deconvolved images is between 140 and 150 nm. Spreading cells were imaged at 2 s intervals and spread cells were imaged at 5 frames per second. The Richardson–Lucy algorithm is used for deconvolution, and run for ten iterations. The PSF used was simulated by a Gaussian function but based on parameters obtained from measurement, i.e., the FWHM of the PSF used is the same as the FWHM measured.

**Data analysis**. The traditional approach for determining diffusion coefficients is to fit the experimental mean-square displacement (MSD) versus delay time to a linear function, yielding the diffusion coefficient as the slope. However, Flyvbjerg et al.[26] showed that this method is inferior to approaches based on the covariance of particle displacements. We find that on an ensemble level, this method yields diffusion coefficients that differ by at most a factor of two from published studies[9]. For individual tracks, even covariance-based methods lead to diffusivities that suffer from significant errors, because of the limited duration of tracks (due to photobleaching), and because measured particle positions are themselves subject to significant errors, both as a result of the limited number of photons from each fluorescent particle and because of the motion blur that inevitably occurs for a nonzero exposure time. In principle, it is possible to mitigate the noise inherent to individual trajectories by averaging over multiple tracks. However, in the heterogeneous cellular environment, the diffusive properties of different trajectories are likely to vary and are unknown a priori. Thus, to employ ensemble averaging, it is necessary to sort trajectories into sub-populations that share diffusive properties. Freeman et al.[15] sought to account for heterogeneity by employing a two-state Hidden Markov model to separate trajectories into high-diffusivity segments and low-diffusivity segments. However, the choice of two diffusive states was imposed by fiat, rather than emerging from the data.

Therefore, in order to obtain a better understanding of the diffusive properties of the BCR, we have employed a newly-introduced methodology, perturbation-expectation maximization (pEM), that sorts a population of trajectories into discrete diffusive states, simultaneously determining the optimal covariance values for each state. Perturbation-expectation maximization version 2 (pEM v2) was used to classify single-molecule tracks derived from different receptors[28]. To perform pEM analysis, all tracks must have the same length. Given the 33 Hz imaging rate, the optimal track length was found to be 15 frames long due to the trade-off between accurately identifying diffusivities and minimizing the number of state transitions that the particle may undergo over a single trajectory[52]. All single-molecule trajectories obtained were split into 15-frame segments and the classification analysis was performed on the set of all these track segments. Trajectories larger than 105 frames or shorter than 15 frames were discarded. The tracking routine interrupts the creation of a trajectory whenever two particles cross paths. To avoid an over counting of slow-moving molecules (which have lower probability of crossing paths with other molecules) we discarded trajectories longer than 105 frames. The data was then separated according to the receptor type and PEM v2 was run for all data sets using 20 reinitializations, 150 perturbations, 14 covariance parameters, and allowing the system to explore up to 15 states. This set of parameter values was chosen to ensure convergence to the global maximum. For all conditions, the average track length was 40 frames and typically 100 tracks were obtained per cell per time point. The maximum posterior probability value was used to assign a track uniquely to a particular state as shown in Supplementary Fig. 9a for BCR in control cells. For BCR, 186,959 tracks corresponding to all inhibitor treatments, DMSO-control and cNKO cells were analyzed together. For CD19, 35,062 tracks corresponding to control and cNKO were pooled together and analyzed, and for FcγRIIB receptor, 24,969 tracks were analyzed. For all the receptors and conditions, eight diffusive states were identified. The states were compared across different receptors based on a comparison of their diffusivity distributions (Supplementary Fig. 9b).

STICS analysis of actin flows was implemented on iSIM images taken at 2s intervals. Subregions of 8 × 8 pixels were selected with a shift of two pixels between subregions. Immobile filtering was set to 20 frames and the time of interest (TOI) was chosen as five frames with a shift of three frames between TOIs. Velocity flow vectors that exceeded the subregion threshold were discarded, giving place to the "black pixels" observed in the decomposed maps of speeds and directions. To determine the directionality of the flow, the centroid of the cell was calculated and a vector from each of the subregions pointing towards the centroid was obtained. The directional coherence was then determined as the cosine of the angle between the velocity vector and the vector pointing to the centroid. Directionality plots were generated using the MATLAB function histcounts and using PDF as normalization type. In order to compare the directionality between DMSO and wiskostatin-treated cells the fraction of inward flow (values larger than 0.9) and the fraction of outward flow (values less than −0.9) was determined. The comparison of fraction of flow in either direction between the two conditions was tested using the z-test where the null hypothesis is that both fractions are equal.

For receptor diffusivity studies 12 control and 9 cNKO mice were used. For actin dynamics studies, three Lifeact-EGFP mice were used.

**Statistical analysis**. The Kruskal–Wallis test was used to assess the difference between the diffusivity distributions corresponding to different conditions. We used this test for most comparisons because it is a nonparametric method for testing whether two data samples originate from the same distribution. The test was performed over smaller data subsets selected randomly and implemented using the Kruskal–Wallis function in MATLAB. The pair wise z-test was used to determine the difference in proportions of diffusive states across different conditions.

**Reporting summary**. Further information on research design is available in the Nature Research Reporting Summary linked to this article.

## Data availability

All data supporting the findings of this study are available from the corresponding author upon reasonable request. Source data has been provided for the following figures: Fig. 2b–e, Fig. 3b, c, e, f, Fig. 4c, d, f, Fig. 5d, f, h, i, Fig. 6c–f, Supplementary Fig. 2b, c, Supplementary Fig. 3, Supplementary Fig. 4b, c, Supplementary Fig. 6d.

## Code availability

The code for pEM analysis is freely available at the following link: https://github.com/p-koo/pEMv2

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

## Acknowledgements

A.U. acknowledges support from the grant NSF PHY 1607645 and NSF PHY 1806903. A.U. and W.S. acknowledge support from the grant NIH R21 AI122205. H.S. acknowledges support from the Intramural Research Programs of the National Institute of Biomedical Imaging and Bioengineering. S.M. acknowledges support by NIGMS AWDA 10958 and the NSF Physics of Living Systems Student Research Network under PHYS 1522467. I.R-S. would like to acknowledge support from the Fulbright-Colciencias scholarship. We would like to thank Dr Peter Tuo Li and Dr Tom Blanpied (UMD, School of Medicine) for guidance on the design of the single-molecule tracking experiments. We thank Harsh Vishwasrao and Jiji Chen of the NIH Advanced Imaging and Microscopy resource for help with iSIM and Kaustubh Wagh (UMD) for useful discussions on actin dynamics analysis. We would like to thank Ethan Tyler (NIH Medical Arts Design Section) for help with illustrations.

## Author contributions

I.R.-S. and A.U. designed the experiments with input from W.S. I.R.-S. performed the experiments and analyzed the data with help from B.A.W. P.K. and S.M. provided guidance on PEM analysis. A.B., Z.S. and W.S. provided guidance and help with spleen and bilayer preparation. H.S. provided guidance on imaging and analysis. I.R.-S. and A.U. wrote the original draft of the paper, with input and editing from H.S., S.M. and W.S. A.U. supervised the project.

## Competing interests

The authors declare no competing interests.
