## [Peer Review File · Nature Communications]

Reviewers' comments:

Reviewer #1 (WASp, actin regulation immune signalling)(Remarks to the Author):

In this work, the authors track single molecules of BCRs, CD19, and FcγRIIB on the B-cell surface following stimulation, and characterize 8 different diffusive states on the B-cell membrane using advanced microscopic and bioinformatics methods. The authors demonstrate that slower diffusive states tend to aggregate into molecule clusters, whereas fast diffusive states occur early in B-cell activation and rapidly decrease in the population fraction. Using N-WASp knockout mice, the authors show that less BCR and CD19 molecules are in fast diffusive states. The authors conclude that N-WASp dictates the diffusive properties of the BCR and CD19, and its absence promotes less actin dynamics, less diffusivity and thus more BCR clustering and activating signaling. The study includes impressive analyses of advanced microscopic data, and progresses our knowledge in the field by elucidating the different dynamics of individual molecules on the B-cell surface following stimulation, and the factors that may dictate B-cell activation threshold. However, the work lacks several critical controls and experiments which must be conducted to support the conclusion of the authors.

1) Arp2/3 is activated through various NPFs including WASp and WAVE 1-3. It is known that WASp and N-WASp have overlapping activities in immune cells. It has already been shown that WASp is dominantly expressed in hematopoietic cells relatively to N-WASp. In light of these data, how do the authors differentiate between the role of WASp and N-WASp in the diffusion of the BCR and CD19? This issue should be carefully addressed. The authors should demonstrate that their experimental system (conditional N-WASp KO mice) and reagents, e.g., inhibitors, etc., are specific to N-WASp. Interestingly, in Figs. 6 and 7 the authors show the effect of wiskostatin on BCR diffusivity. Again, this inhibitor down regulates WASp activity, and thus the effects the authors demonstrate might be through WASp. This issue should be resolved.

2) In Fig. 6 the authors show that formin inhibition also affects BCR diffusion. Formins generate completely different actin structures and facilitate actin polymerization differently and independently from WASp family proteins. Thus, it appears that inhibiting actin polymerization from different pathways affects diffusivity, and not just through N-WASp. This issue should be addressed by the authors.

3) Furthermore, N-WASp has several activities besides actin nucleation, including facilitating transcription. In order to prove that the nucleation activity of N-WASp directly impacts BCR and CD19 diffusivity, the authors should use an N-WASp mutant in the VCA domain that cannot facilitate actin nucleation.

4) Fig. 6 lacks the effect of actin dynamics on the diffusivity of CD19 and FcγRIIB that were shown to be dependent on N-WASp by the authors. The authors should also show the effect of actin nucleation (as shown by the usage of inhibitors in Fig.6) on the diffusivity of CD19 and FcγRIIB, and not only on the BCR.

5) The whole study was conducted in a murine model. It will be informative if diffusion analysis and inhibition of N-WASp activity will be performed also on human cells. It is important to emphasize the point that mice deficient in WASp do not seem to display the same severe phenotypes as demonstrated in humans. While WASp deficiency is lethal in human (patients that completely lack WASp die within the first decade of their lives if not treated), the mice are viable. It might be that the effects seen by the authors for murine B-cells are unique to mice and do not reflect the human system.

6) Fig. 1 and Fig S1 lack controls of panels showing B-cell spreading on irrelevant antibody. This experiment should be performed to demonstrate diffusivity under basal conditions and to prove that activation occurred due to BCR stimulation. Moreover, the diffusive states of BCRs and CD19 on non-stimulated B-cells should be demonstrated. Do naïve B cells demonstrate the same number of diffusive states, in which the fastest state decreases rapidly over-time, or is this a direct function of B-cell activation? Is this also true for FcγRIIB mobility?

7) Figure 7e is wrongly labeled as 7h.

Reviewer #2 (B cell activation, BCR signalling)(Remarks to the Author):

The manuscript by Suarez et al investigated the function of actin dynamics on the mobility of BCR and other co-receptors (CD19 and CD32) during B cell signaling. An attractive flavor of this study is that authors developed a novel machine learning based method to classify BCR trajectories into distinct diffusive states. By doing so, the authors revealed that the actin regulatory protein N-WASP potently regulated receptor mobility. Loss of N-WASP led to a predominance of BCR trajectories with lower diffusivity that is correlated with a decrease in actin dynamics and enhanced BCR signaling. Another novel finding by this study is that loss of N-WASP reduced diffusivity of BCR activating co-receptor, CD19, but not that of BCR inhibitory co-receptor, FcγRIIB. Overall, this manuscript revealed how actin dynamics regulated the nanoscale organization and diffusivity of BCR and its co-receptors during B cell activation, which shall be of general interest to plasma membrane and receptor biologists. However the authors shall address the following questions before the consideration of the publication of this manuscript.

- 1: The authors shall significantly demonstrate the advantages and disadvantages of the newly established machine learning based method to analyze BCR trajectories since this shall be the core novelty of this manuscript. There are some published studies similarly working on BCR mobility on both quiescent and activated B cells. The authors shall explicitly reveal the novelty of their method.
- 2: The No.1 point is especially important as BCR mobility in this study (Fig. 1f) is much higher than those in the published studies. Although these absolute values can vary due to different experimental conditions and MSD-based trajectory calculations, the authors shall at least thoroughly discuss this point and ideally provide the mobility measurement of lipid molecules in planar lipid bilayers in the imaging acquisition and analysis condition in this report as the mobility of these standard molecules is more stable and can be compared.
- 3: The reduction in BCR diffusivity was observed in N-WASP KO B cells and B cells pre-treated by CK666 (Arp2/3 complex inhibitor) or SMIF2 (formin inhibitor) inhibitors. Based on these findings, the authors made the core conclusion that N-WASP regulated BCR mobility through Arp2/3 complex. This reviewer would comment that this is a conclusion lack of solid evidence. Although the vital role played by N-WASP in regulating actin polymerization by stimulating the actin-nucleating activity of the Arp2/3 complex have been widely reported, other function of N-WASP or potential side-effect of pharmacology treatment should be taken into consideration. The observed similarity of different experiment systems will not be used as solid evidence for signal axis transduction in a given physiology process. The authors shall provide more direct evidence to validate the hypothesis that the heterogeneity of BCR mobility was maintained or controlled by actin dynamic via N-WASP.
- 4: The authors shall thoroughly discuss the new findings in this report in comparison to their published studies in 2013 showing that N-wasp is essential for the negative regulation of B cell receptor signaling.
- 5: In general, the authors shall avoid a repetition of results that were previously published even if they are presented in other ways. For example, the statement of Actin or N-WASP engagement in BCR mobility and signal transduction has been described too many times in previous papers and in this manuscript. There are many places in the manuscript that shall be revised. I just listed some of them: page 12 line 370-372; page 13 line 388-390; page 13 line 397-399.
- 6: The novelty of "Actin barrier model" showed in Fig.8 shall be thoroughly discussed since there are some studies reporting the function of F-actin filaments in regulating BCR mobility in the literature studies.

Reviewer #1 (Remarks to the Author):

1) Arp2/3 is activated through various NPFs including WASp and WAVE 1-3. It is known that WASp and N-WASp have overlapping activities in immune cells. It has already been shown that WASp is dominantly expressed in hematopoietic cells relatively to N-WASp. In light of these data, how do the authors differentiate between the role of WASp and N-WASp in the diffusion of the BCR and CD19? This issue should be carefully addressed.

We appreciate the reviewer's suggestion about considering the effects of WASP on BCR diffusion and their similarity or differences compared to N-WASP. We note that at the mRNA level, Raji B cells only show a 30% difference between WASP and N-WASP (Jain and Thanabalu, *Sci. Rep.* 5:15031, 2015). Moreover, N-WASP is insensitive to cleavage by the protease, calpain (Scherbina et al. *Blood* 98:2988, 2011). Thus, at the protein level, the abundance of these two regulators might be similar. As shown in our prior work (Liu et al. *PLoS Biol* 2013), *both* N-WASP and WASP are required for antigen gathering and the dynamics of BCR signaling. In contrast, B cells from WASP-KO (WKO) mice behave differently compared to those from B cell specific conditional N-WASP-KO (cNKO) cells. While antigen gathering is reduced both in WKO and cNKO B cells, WKO cells spread less and have lower signaling levels and cNKO B cells spread more and have higher signaling levels, compared to control B cells. This would imply that WKO and cNKO could affect BCR diffusion/mobility differently.

To test this hypothesis, we compared the diffusion of BCRs on the surface of splenic B cells from cNKO and WKO mice using single molecule techniques. We found that loss of either N-WASP or WASP results in a reduction of nanoscale BCR diffusivity, but to a lesser degree in WKO B cells as compared to cNKO B cells. Further, perturbation Expectation Maximization (pEM) analysis of the single molecule tracks revealed significant changes in the diffusive states in both KO cells compared to control. Notably, State 1 in WKO cells had a similar population fraction as in control cells, while State 2 was modestly increased. However, in cNKO cells both State 1 and State 2 had significantly higher population fractions compared to control cells. This indicates that WASP and N-WASP have differential effects on the mobility and putative signaling states of BCR. We have now added these data and analyses in the revised manuscript in Fig. 4e-f and Supplementary Fig. S4.

The authors should demonstrate that their experimental system (conditional N-WASp KO mice) and reagents, e.g., inhibitors, etc., are specific to N-WASp. Interestingly, in Figs. 6 and 7 the authors show the effect of wiskostatin on BCR diffusivity. Again, this inhibitor down regulates WASp activity, and thus the effects the authors demonstrate might be through WASp. This issue should be resolved.

Both WASP KO and conditional N-WASP KO mice have been established and extensively characterized by the Snapper lab. Specifically, in our previous published paper, we showed the efficient deletion of N-WASP in the conditional N-WASP knockout and efficient deletion of WASP in the WASP knockout mice at both the mRNA and protein levels (Liu et al., *PLoS Biol.* 2013). Furthermore, the expression levels of WASP in cNKO B cells were unchanged from that in B cells from littermate controls. Similarly, the expression levels of N-WASP in the WKO B cells were similar to B cells from littermate controls (Liu et al., *PLoS Biol.* 2013, Supplementary Figures 3 and 5).

While Wiskotatin has been shown to bind both WASP and N-WASP in an in vitro reconstituted system (Peterson et al. Nat. Cell Biol. 2004), we have shown in our previous publication (Liu et al. PLoS Biol. 2013) that Wiskostatin treatment inhibits the phosphorylation of N-WASP but enhances the phosphorylation of WASP. The effect of Wiskostatin treatment on B cell spreading, antigen gathering and WASP phosphorylation is similar to what is observed in cNKO B cells. These data suggest that the function of Wiskostatin is likely context/cell-type specific. However, in the manuscript we have removed any suggestions that wiskostatin is specific for N-WASP.

However, given our new studies with WASP KO B cells along with the observations for cNKO B cells, we have now revised the manuscript to emphasize that actin dynamics through multiple pathways plays a key role in modulating nanoscale BCR mobility. Moreover, our studies with B cells from cNKO mice allow us to demonstrate that BCR diffusivity is correlated with its signaling state. Thus, demonstrating that our observed effects are specific to N-WASP is no longer central to our manuscript.

2) In Fig. 6 the authors show that formin inhibition also affects BCR diffusion. Formins generate completely different actin structures and facilitate actin polymerization differently and independently from WASp family proteins. Thus, it appears that inhibiting actin polymerization from different pathways affects diffusivity, and not just through N-WASp. This issue should be addressed by the authors.

We have now addressed this point in the revised manuscript (Pg. 14). As noted by the reviewer, in addition to the branched actin nucleator Arp2/3 (downstream of WASP and N-WASP), formin also regulates BCR diffusion. The effect of formin inhibition may be due either to direct alterations of linear actin structures produced by formin or by alteration of the branched actin networks produced by the cooperation of Arp2/3 and Diaphanous formins. Such cooperative interactions have been observed between the formin mDia1 and Arp2/3 for the generation of lamellipodia and ruffles on HeLa cells (Isogai et al. J. Cell Sci. 2015) and the formation of proper actin architecture at the fusion site during invasive podosome formation in fruit flies (Deng et al. PLoS Genetics 2015). Furthermore, T cells obtained from mDia1-knockout mice cannot form lamellipodia or ruffles, and exhibit defective cell motility (Sakata et al., J. Exp. Med. 2007). Moreover, the polymerization of branched filaments by Arp2/3 requires existing actin filaments. Formins can generate the mother filaments for Arp2/3 to form the branched actin meshwork that underlies the formation of different types of cell protrusions including B cell spreading.

3) Furthermore, N-WASp has several activities besides actin nucleation, including facilitating transcription. In order to prove that the nucleation activity of N-WASp directly impacts BCR and CD19 diffusivity, the authors should use an N-WASp mutant in the VCA domain that cannot facilitate actin nucleation.

We recognize this important point raised by the reviewer. In our prior study, we carefully compared the expression levels of WASP and N-WASP in cNKO and WKO cells respectively (see Supplementary Fig. 5 in Liu et al). We found that the expression levels of WASP in cNKO cells were similar to expression levels in littermate controls. Also, the expression levels of N-WASP in WKO cells were unchanged compared to littermate controls. This indicates that gene KO does not affect the overall expression levels of N-WASP and WASP, suggesting that the KO does not lead to off-target effects. We agree that N-WASP with only the VCA domain mutated may preserve the

other functions of N-WASP. However, generation of B cell-specific knock-in mice with mutations in the VCA domain of N-WASP will take beyond a reasonable time window for manuscript revision. In addition, as the VCA domain of N-WASP is also required for the inhibitory conformation of N-WASP, any mutation in the VCA domain may change the functional state of N-WASP.

4) *Fig. 6 lacks the effect of actin dynamics on the diffusivity of CD19 and FcγRIIB that were shown to be dependent on N-WASp by the authors. The authors should also show the effect of actin nucleation (as shown by the usage of inhibitors in Fig.6) on the diffusivity of CD19 and FcγRIIB, and not only on the BCR.*

To address the reviewer's comments, we examined the effect of Wiskostatin (10 μM), SMIFH2 (25 μM) and CK666 (50 μM) on CD19 diffusivity in independent experiments (2 different mice at 2 different times) and measured CD19 diffusivity using single molecule imaging. For every combination of inhibitor and receptor, 10 or more cells were imaged for at least 10 minutes under conditions appropriate for single molecule imaging. These inhibitors caused various levels of reduction in CD19 diffusivity when compared to the control. For all cases, there is a significant increase in population fractions of states 1, 3 and 6. These new data are added as Supplementary Figure S6c-d in the revised manuscript. We also studied the effect of the inhibitors on the diffusivity of FcγRIIB. Consistent with our findings in cNKO B cells, addition of Wiskostatin, SMIFH2 or CK666 did not alter the overall diffusivity or the population fraction of diffusive states of FcγRIIB.

5) *The whole study was conducted in a murine model. It will be informative if diffusion analysis and inhibition of N-WASp activity will be performed also on human cells. It is important to emphasize the point that mice deficient in WASp do not seem to display the same severe phenotypes as demonstrated in humans. While WASp deficiency is lethal in human (patients that completely lack WASp die within the first decade of their lives if not treated), the mice are viable. It might be that the effects seen by the authors for murine B-cells are unique to mice and do not reflect the human system.*

We agree that primary human cells will make this study more complete. However, we have previously showed (Liu et al. PLoS Biol. 2013) human B cells from Wiskott-Aldrich symptom patients and healthy controls behave similarly as B cells from WKO and littermate control mice, including similar kinetics of WASP and N-WASP phosphorylation and similar effects of wiskostatin treatment. Wiskostatin-treated primary human B cells exhibited enhanced spreading and reduced antigen gathering similar to B cells from cNKO mice. These results suggest that primary cells from mice and human would exhibit similar results for nanoscale BCR diffusivity.

According to the literature, WAS patients die in young age from repeated bacterial, viral, and fungal infections due to immune deficiency, partially caused by defected B cell functions. WKO mice are viable for longer time, since they have been kept in specific pathogen-free environment, avoiding of infections that are lethal to WKO mice. WKO mice has been widely used to study WAS disease, as it is a rare disease in humans.

Furthermore, it is technically impractical for us to examine human B cells for the revision of this manuscript, as we need a significant amount of time to get an IRB protocol approved, identify sources of human peripheral blood cells, and establish treatment and imaging conditions. We

believe that a lack of human cell data does not diminish the significance of our findings, given that our published studies established that human B cells behave similarly to mouse B cells in the context of the effect of WASP and N-WASP on BCR accumulation.

6) Fig. 1 and Fig S1 lack controls of panels showing B-cell spreading on irrelevant antibody. This experiment should be performed to demonstrate diffusivity under basal conditions and to prove that activation occurred due to BCR stimulation. Moreover, the diffusive states of BCRs and CD19 on non-stimulated B-cells should be demonstrated. Do naïve B cells demonstrate the same number of diffusive states, in which the fastest state decreases rapidly over-time, or is this a direct function of B-cell activation? Is this also true for FcγRIIB mobility?

7) Figure 7e is wrongly labeled as 7h.

We thank the reviewer for these suggestions and points. We have performed single molecule imaging of BCR in murine B cells on transferrin-tethered lipid bilayers as non-activating control. B cells bind to transferrin-tethered lipid bilayers through transferrin receptor, but the binding does not activate the BCR. We found that BCR diffusivity of B cells on transferrin lipid bilayers is higher and did not show a progressive decrease over time in contrast to B cells on an activating bilayer. Moreover, pEM analysis of BCR diffusivities showed a near complete loss of the population of trajectories in the lowest mobility states (1 and 2). These data are now shown in Supplementary Figure 2. We have also fixed the labeling in Fig. 7.

Reviewer #2

The manuscript by Suarez et al investigated the function of actin dynamics on the mobility of BCR and other co-receptors (CD19 and CD32) during B cell signaling. An attractive flavor of this study is that authors developed a novel machine learning based method to classify BCR trajectories into distinct diffusive states. By doing so, the authors revealed that the actin regulatory protein N-WASP potently regulated receptor mobility. Loss of N-WASP led to a predominance of BCR trajectories with lower diffusivity that is correlated with a decrease in actin dynamics and enhanced BCR signaling. Another novel finding by this study is that loss of N-WASP reduced diffusivity of BCR activating co-receptor, CD19, but not that of BCR inhibitory co-receptor, FcγRIIB. Overall, this manuscript revealed how actin dynamics regulated the nanoscale organization and diffusivity of BCR and its co-receptors during B cell activation, which shall be of general interest to plasma membrane and receptor biologists. However the authors shall address the following questions before the consideration of the publication of this manuscript.

1: The authors shall significantly demonstrate the advantages and disadvantages of the newly established machine learning based method to analyze BCR trajectories since this shall be the core novelty of this manuscript. There are some published studies similarly working on BCR mobility on both quiescent and activated B cells. The authors shall explicitly reveal the novelty of their method.

We thank the reviewer for this comment and have now explicitly discussed the novelty of our method in comparison to previous methods. In brief, pEM analysis of all BCR trajectories from WT identified 8 distinct states, revealing a far greater complexity of diffusive behavior than is apparent from approaches that average over all tracks or that impose two diffusive states only. We have included this in the Results section (pg. 5) and Methods (pg. 18) of our revised manuscript.

2: The No.1 point is especially important as BCR mobility in this study (Fig. 1f) is much higher than those in the published studies. Although these absolute values can vary due to different experimental conditions and MSD-based trajectory calculations, the authors shall at least thoroughly discuss this point and ideally provide the mobility measurement of lipid molecules in planar lipid bilayers in the imaging acquisition and analysis condition in this report as the mobility of these standard molecules is more stable and can be compared.

In order to address this point, we decided to compare our data with the MSD (Mean Square Displacement) plot published by Tolar et al. (Immunity 2009. Figure S2). The MSD is constructed from the particle tracks directly using the formula $MSD(T) = \langle [r(t+T) - r(t)]^2 \rangle$, where $r(t)$ is the position of the particle at time t , and T is the lag time between the two positions of the particle used to calculate the displacement. The $\langle \rangle$ represents a time average over t and/or an ensemble average over multiple tracks. To obtain the diffusivity of BCR (and the other receptors we studied), we used the covariance-based estimation method developed by Vestergaard et al. (Vestergaard CL, Blainey PC, Flyvbjerg H, 2014, Phys Rev E 89:22726.). The covariance-based estimator (CVE) is unbiased and does not need a regression analysis to estimate diffusion coefficients. Therefore, this method is ideal for obtaining diffusion coefficients from short single particle trajectories. The plot below shows an overlay of the data points from the MSD for BCR of B cells activated with monovalent antigen (NIP-H12) from Tolar et al. with our data. The overlaid plot shows that both curves have the same slope (with different Y-intercepts possibly due to a difference in acquisition equipment). Thus the difference in the diffusivities between our manuscript and previous reports arises from the method used to calculate the diffusivity but the differences are at most a factor of 2. We have remarked upon this point in the Methods section (pg 18).

Comparison of estimated Mean Squared Displacements for BCR-mbFab (this manuscript) and for NIP-H12 by Tolar et al. The slopes of the lines are very similar indicating similar diffusion constants to that calculated by our method.

3: The reduction in BCR diffusivity was observed in N-WASP KO B cells and B cells pre-treated by CK666 (Arp2/3 complex inhibitor) or SMIFH2 (formin inhibitor) inhibitors. Based on these findings, the authors made the core conclusion that N-WASP regulated BCR mobility through Arp2/3 complex. This reviewer would comment that this is a conclusion lack of solid evidence. Although the vital role played by N-WASP in regulating actin polymerization by stimulating the actin-nucleating activity of the Arp2/3 complex have been widely reported, other function of N-WASP or potential side-effect of pharmacology treatment should be taken into consideration. The observed similarity of different experiment systems will not be used as solid evidence for signal axis transduction in a given physiology process. The authors shall provide more direct evidence to

validate the hypothesis that the heterogeneity of BCR mobility was maintained or controlled by actin dynamic via N-WASP.

We agree that our data only showed a correlation between the N-WASP gene knockout and Arp2/3 inhibitor on BCR diffusivity but did not provide a direct link between N-WASP and Arp2/3 for BCR diffusivity. Unfortunately, we are unable to provide such direct evidence due to technical difficulties in imaging both actin and BCR at the single molecule level as well as generating knockin mice with mutations of various N-WASP actin interacting domains. In the revised manuscript, we have modified our conclusions to place more emphasis on the connection between actin dynamics and the heterogeneity of BCR mobility. The conditional N-WASP knockout provides a way to relate BCR signaling and diffusivity. However, based on our inhibition experiments and our prior work, our findings strongly suggest that N-WASP acts to modulate BCR nanoscale mobility and signaling through its effects on actin dynamics.

4: The authors shall thoroughly discuss the new findings in this report in comparison to their published studies in 2013 showing that N-wasp is essential for the negative regulation of B cell receptor signaling.

We have now amended the discussion section (pg. 13) to highlight the novel results in our manuscript.

5: In general, the authors shall avoid a repetition of results that were previously published even if they are presented in other ways. For example, the statement of Actin or N-WASP engagement in BCR mobility and signal transduction has been described too many times in previous papers and in this manuscript. There are many places in the manuscript that shall be revised. I just listed some of them: page 12 line 370-372; page 13 line 388-390; page 13 line 397-399.

We have revised the manuscript to remove any redundant statements.

6: The novelty of “Actin barrier model” showed in Fig.8 shall be thoroughly discussed since there are some studies reporting the function of F-actin filaments in regulating BCR mobility in the literature studies.

We note that our model goes beyond the well-accepted role of actin cytoskeleton as barriers for BCR diffusion. We suggest, based on our observations, that the dynamic actin cytoskeleton may serve to serve to ‘stir’ the cytoplasm adjacent to the membrane, thus changing the reaction environment of receptors and signaling molecules, thereby regulating the nanoscale mobility and spatial organization of the BCR. We now highlight this in our revised discussion (pg. 14-15) and contrast our new model with previously proposed ones.

REVIEWERS' COMMENTS:

Reviewer #1 (Remarks to the Author):

The authors properly addressed my concerns.

Reviewer #2 (Remarks to the Author):

In the revised version of this manuscript, the authors fully addressed the questions that I have raised to their original submission. I do not have further concerns and would recommend the publication of this paper.

REVIEWERS' COMMENTS:

Reviewer #1 (Remarks to the Author):

The authors properly addressed my concerns.

Response: We are pleased that the Reviewer is satisfied with our revised manuscript.

Reviewer #2 (Remarks to the Author):

In the revised version of this manuscript, the authors fully addressed the questions that I have raised to their original submission. I do not have further concerns and would recommend the publication of this paper.

Response: We are pleased that the Reviewer is satisfied with our revised manuscript.